# Clinical Outcome of Coronavirus Disease 2019 in Patients with Primary Antibody Deficiencies

**DOI:** 10.3390/pathogens12010109

**Published:** 2023-01-09

**Authors:** Tomas Milota, Jitka Smetanova, Jirina Bartunkova

**Affiliations:** Department of Immunology, Second Faculty of Medicine Charles University and Motol University Hospital, 15006 Prague, Czech Republic

**Keywords:** COVID-19, SARS-CoV-2, primary antibody deficiency

## Abstract

In 2019, the novel coronavirus, SARS-CoV-2, caused a worldwide pandemic, affecting more than 630 million individuals and causing 6.5 million deaths. In the general population, poorer outcomes have been associated with older age, chronic lung and cardiovascular diseases, and lymphopenia, highlighting the important role of cellular immunity in the immune response against SARS-CoV-2. Moreover, SARS-CoV-2 variants may have a significant impact on disease severity. There is a significant overlap with complications commonly found in inborn errors of immunity (IEI), such as primary antibody deficiencies. The results of various studies have provided ambiguous findings. Several studies identified risk factors in the general population with a minor impact on SARS-CoV-2 infection. However, other studies have found a significant contribution of underlying immunodeficiency and immune-system dysregulation to the disease course. This ambiguity probably reflects the demographic differences and viral evolution. Impaired antibody production was associated with prolonged viral shedding, suggesting a critical role of humoral immunity in controlling SARS-CoV-2 infection. This may explain the poorer outcomes in primary antibody deficiencies compared to other IEIs. Understanding coronavirus disease 2019 (COVID-19) pathogenesis and identifying risk factors may help us identify patients at high risk of severe COVID-19 for whom preventive measures should be introduced.

## 1. Introduction

Coronaviruses are a family of RNA viruses that cause mild respiratory infections. The perception of coronaviruses changed in 2002 when the first SARS-CoV-positive patient was reported in the Chinese city of Foshan [1]. Seventeen years later, the first cases of severe respiratory disorders similar to those caused by severe acute respiratory syndrome coronavirus (SARS-CoV-2) were reported in the Chinese city of Wuhan [2]. The World Health Organization has reported more than 630 million cases and 6.5 million deaths to date [3]. Older age and chronic lung and cardiovascular diseases are the main risk factors for poor outcomes. This spectrum of complications overlaps with the noninfectious manifestations of primary antibody deficiencies (PADs) [4,5]. Moreover, an increased risk of a poorer outcome was also found in patients with systemic autoimmune diseases [6] or after hematopoietic cell and solid-organ transplantation [7]. These observations have raised questions regarding the impact of coronavirus disease 2019 (COVID-19) on patients with inborn errors of immunity (IEIs). This overview provides complex insights into the clinical outcomes of COVID-19 in patients with PADs. We discuss this in the context of viral evolution, the pathogenesis of infection, and risk factors for poor outcomes. The review was prepared in line with the Preferred Reporting Items for Systematic Reviews and Meta-Analyses guidelines [8] and the proposed guidelines for biomedical narrative review [9].

## 2. Primary Antibody Deficiency

PADs are a heterogeneous group of inborn errors of immunity (IEI). PADs represent more than half of all cases of IEIs [10,11,12]. The spectrum of PADs ranges from selective deficiencies (selective IgA deficiency, selective IgM deficiency, IgG1–IgG4 subclass deficiencies) and impaired production of specific antibodies to severe and complex immunodeficiencies, such as common variable immunodeficiency (CVID) or X-linked agammaglobulinemia (XLA). However, disturbed production of antibodies may also accompany other groups of IEIs, such as combined immunodeficiencies or diseases of immune dysregulation [13]. Selective IgA deficiency (sIgAD) and common variable immunodeficiency (CVID) belong to the most-frequent PADs, with an incidence of 1:1000–1:140 [14] and 0.08–3.14:100 000 in Europe and 1.48 in the USA [15]. Pathophysiologically, impaired maturation of B cells is the hallmark of most PADs. Generally, most PADs are multifactorial diseases with polygenic inheritance. To date, 45 genes have been identified as monogenic causes of PADs [16]. The genetic background is known in inherited forms of agammaglobulinemia. Mutations in the X-linked BTK gene encoding Bruton’s tyrosine kinase are the most-common cause of inherited agammaglobulinemia. More than 600 mutations have been described to date, 10–15% of which occur de novo. Most of them involve 1–4 base pairs. However, larger deletions have been found in 3–5% of patients. The deleted regions may damage other linked or adjacent genes, such as TIMM8A and TAF7L, resulting in XLA and deafness-dystonia-optic neuropathy syndrome. BTK is essential for BCR-mediated proliferation, survival, and B-cell development. BTK transduces signals from pre-BCR complexes. Autosomal dominant (LRRC8, TOP2B) and recessive forms (μ heavy chain, Lambda 5, Ig alpha/beta, BLNK, PI3K genes, and TCF3) represent rare forms of inherited agammaglobulinemia. Most of them affect the pre-B-cell/B-cell receptor complex or signaling pathways, and [17,18,19] monogenic causes of CVID have been found in approximately 10% of patients. Mutations in PIK3CD, PIK3R1, NFkB1/2, CTLA4, and LRBA are the most clinically relevant genes [20,21,22]. Similarly, mutations in JAK3, RAG1, DCLRE1C, CD27, LRBA, BTK, TACI, TWEAK, MSH6, MSH2, PIK3R1, and CARD11 have been identified as underlying gene defects in sIgAD [23]. While the developmental block of B cells in inherited agammaglobulinemia occurs at the pre-B-cell stage, it predominantly affects memory cells in sIgAD and CVID. Several studies have reported decreased class-switched memory B cells and plasmablasts in patients with sIgAD and CVID. On the other hand, elevated counts of CD21 (low) and CD38 (low) B cells reflect immune-system dysregulation and are found in patients with splenomegaly and granulomatous complications [24,25,26]. They have both proinflammatory and autoreactive properties [27,28,29]. A reduced number of B-regulatory cells [30], T-cell abnormalities, such as reduced Treg counts, increased T-cell activation, apoptosis, and exhaustion [31,32], may further contribute to immune-system dysregulation. The frequency of Tregs is negatively correlated with the counts of CD21(low) and CD38(low) B cells [33]. Severely reduced CD4+ T cells are found in specific subgroups of CVID, late-onset combined immunodeficiency (LOCID), and have a higher frequency of splenomegaly, granuloma, GI disease, and lymphoma [34].

PADs are characterized by increased susceptibility, particularly to respiratory tract bacterial infections, usually caused by *Streptococcus pneumoniae* or *Haemophilus influenzae*. However, other infections, mainly those affecting the gastrointestinal tract, are also frequent, such as *Giardia lamblia*, salmonella, and campylobacter spp. [35]. There is also a clinically significant increase in susceptibility to viral infections caused by enteroviruses and noroviruses [36]. Noninfectious complications represent other features that significantly contribute to the morbidity and mortality of patients with PADs. The spectrum includes hematologic and organ-specific autoimmune diseases, chronic lung disease, enteropathies mimicking inflammatory bowel diseases or celiac disease, granulomas, and lymphoproliferative disorders, such as lymphadenopathy, splenomegaly, or nodular lymphoid hyperplasia [4,5]. Non-infectious manifestations of CVID has been also summarized in meta-analysis by Janssen et al. (Table 1) [37].

The spectrum and severity of infectious and noninfectious complications is reflected in the therapeutic approach. Essentially, the treatment is based on immunoglobulin replacement (IRT) with intravenous or subcutaneous immunoglobulins, which is indicated in severe hypogammaglobulinemia, disturbed specific antibody immune response, and severe recurrent infections, with or without concomitant antibiotic treatment for the prevention of infectious complications [38]. Currently, the recommended starting IRT dose is 400–600 mg/kg per month to reach serum IgG trough level 7–8 g/L. The dose of IRT should be considered individually to minimize the number of acute infections. Higher doses are required, particularly in patients with bronchiectasis or severe noninfectious complications [39]. Noninfectious complications require immunosuppressive treatment that is almost the same as in the patients with primary autoimmune diseases [40,41], including glucocorticoids and conventional synthetic disease-modifying drugs (such as sulfasalazine, cyclosporine, or methotrexate) [42,43]. Targeted therapy is available for gene-specific diseases, such as abatacept for CTLA4 deficiency [44,45] and mTOR or PI3K inhibitors in APDS [46,47]. As B cells play a crucial role in the pathogenesis of noninfectious complications, rituximab (an anti-CD20 chimeric monoclonal antibody) has proven to be an effective and safe therapeutic option in the treatment of various noninfectious complications as a second-line treatment [48]. Immunosuppression deepens the immunodeficiency state and increases the risk of poorer outcomes of COVID-19, as discussed later.

## 3. Origin of SARS-CoV-2

Understanding the origin, structure, and functional properties of SARS-CoV-2 and the pathogenesis of COVID-19 has important implications for therapy and diagnostics. Coronaviruses are a family of enveloped RNA viruses, consisting of four genera: alpha-, beta-, gamma-, and deltacoronaviruses. Alpha- and betacoronaviruses infect only mammals and usually cause respiratory diseases in humans and gastrointestinal infections in animals. While four human coronaviruses—HcoV-NL63, HCoV-229E, HCoV-OC43, and HKU1—cause mild upper respiratory tract infections in immunocompetent individuals, there are three highly pathogenic variants—MERS-CoV, SARS-CoV, and SARS-CoV-2—associated with Middle East Respiratory Syndrome (MERS), Severe Acute Respiratory Syndrome (SARS), and Coronavirus Disease 2019 (COVID-19), respectively [1]. The first case of SARS-CoV infection was described in 2002 in the Chinese city of Foshan. The infection spread to 27 countries worldwide until July 2003, when no more infections were detected. During the pandemic, SARS infected more than 8000 patients and caused almost 800 deaths, corresponding to nearly 10% mortality. Ten years later, a male patient died from acute pneumonia and renal failure in 2012. A novel coronavirus, MERS-CoV, was detected in the patient’s sputum. Until 2016, more than 1700 confirmed cases, including 624 deaths (35% mortality), were reported. The sources of infections showed civets, raccoon dogs, and ferret badgers as potential reservoirs for SARS-CoV and dromedary camels for MERS-CoV [49].

At the end of 2019, a new member of the coronavirus family, SARS-CoV-2, caused a worldwide outbreak of unusual pneumonia. The first patient was reported in the Chinese city of Wuhan, China. SARS-CoV-2 showed similarities to bat and pangolin coronaviruses, suggesting an evolutionary relationship rather than a viral source [50]. Contact tracing of SARS-CoV-2 to Wuhan animal markets showed similarities to SARS-CoV, but a specific reservoir has not yet been identified [51]. Transmission from humans to animals has been documented in minks, domestic cats, and dogs. Nevertheless, the relevance of infection in domestic pets seems to be limited, as there is no evidence of transmission back to humans or other species. However, minks may be at a high risk of continued re-infection. Further investigation, vigilance, and screening are required to prevent circulation and eliminate potential reservoirs [52].

## 4. Structure and Pathogenesis of SARS-CoV-2

The S, E, and N proteins are essential viral structures (Figure 1). S protein is a highly glycosylated membrane protein. It mediates the fusion of the viral membrane with the host cell membrane. It consists of the S1 and S2 subunits. The S1 subunit contains receptor-binding domains (RBDs) and N-terminal domains (NTDs). The RBD recognizes type 2 angiotensin-converting enzyme (ACE2), which binds to sugar motifs after initial attachment. The S2 unit, with the structure of the ion channel, regulates the fusion of the viral membrane and drives lysis and viral genome release. Moreover, the E protein is involved in viral assembly and transfer to the endoplasmic reticulum and Golgi body membranes. The N protein has two structurally conserved domains, NTD and CTD, which exist as monomers (NTD) or dimers (CTD). N protein is a crucial component that protects the viral genome. The ribonucleoprotein complex comprises the N protein and viral RNA. Additionally, it can serve as a viral suppressor of RNA, thereby mitigating the host antiviral response. The main protease and RNA-dependent RNA polymerase play critical roles in the replication of SARS-CoV-2 [53,54]. There are several sites for virus entry, including nasal, oral ocular, and respiratory, which are rich in ACE2 expression.

Although SARS-CoV-2 was initially thought to be a respiratory disease, it may have disseminated and evolved into multiorgan infections. The positive detection of viremia was found in 28–32% of hospitalized patients and increased to 78% in patients requiring intensive care. Cardiovascular, liver, kidney, and pancreatic involvement have been reported [55]. Symptoms usually develop within the first day of infection. The viral load peaked in the first week. The early peak within the first days of symptoms can explain the higher infectiousness compared to SARS-CoV-1. SARS-CoV-2 RNA can be detected for a mean of 17 days after symptom onset. Prolonged viral shedding has been reported, particularly in immunocompromised or severely ill patients [56]. However, infectiousness is also dependent on the virus variant. After entry into cells, viral RNA is detected by cytoplasmic pattern recognition receptors, such as MDA5 or endosomal Toll-like receptors (TLR 3, TLR7/8), that initiate the expression of type I and III interferons and other pro-inflammatory cytokines and chemokines [57,58,59]. In most patients, the initial immune response can control disease progression. However, patients with severe COVID-19, for example, patients with hypoxic respiratory failure, develop hyperinflammation with the release of pro-inflammatory cytokines, such as interleukin (IL)-1, IL-6, IL-8, or Tumor necrosis factor alpha (TNFα). In particular, serum levels of IL-6, IL-8, and TNFα at the time of hospital admission were strong predictors of patient survival. Severe disease is also associated with elevated inflammatory markers—C-reactive protein (CRP), ferritin, and D dimers [60]. Subsequently, professional antigen-presenting cell (APC)–dendritic cells and macrophages present viral peptides to CD4+ T-helper cells through MHC class II molecules that interact with B cells. APC activates CD8+ T cytotoxic cells through MHC-class I. T lymphocytes then develop into effector and memory cells. Cytotoxic T cells represent one of the main effector mechanisms that lead to the lysis of infected cells. B cells directly recognize viral antigens and produce an IgM isotype within the first week following symptoms [61,62]. The immune response against SARS-CoV-2 is summarized in Figure 2. Humoral responses are elicited against N and S proteins. Peak titers and neutralizing antibodies were observed at 2–3 weeks. IgM levels began to decline after 1–3 months. Specific IgG antibodies may persist for up to one year. A higher antibody response has been seen in severe COVID-19 compared to mild disease, probably due to the high viral load in the severe course associated with a stronger immune response [63,65]. Similarly, the T-cell response has been observed against almost all viral proteins. CD4+ T cells are correlated with the level and persistence of antibody responses. T-cell immunity may be maintained for up to 10–12 months. The response level depends on the clinical severity of the infection. T-cell responses have been detected in individuals who failed to seroconvert after asymptomatic or mild COVID-19. While humoral immunity is important during the first phase of infection for virus neutralization, a cytotoxic cellular response is required for viral clearance. The elimination of infected cells prevents virus release and spread resistant to antibody neutralization. The importance of cellular immunity is highlighted by lymphopenia, which is associated with poor clinical outcomes. However, a severe clinical course may be mediated by excessive T-cell activation, contributing to tissue damage [66,67,68]. The efficacy of anti-SARS-CoV-2 antibodies is also dependent on virus variants. Antibodies induced by the wild-type variant could cross-neutralize the beta variant (B.1.351) but with 2–4-times lower neutralizing capacity. Interestingly, no differences were observed in CD4+ T-cell activation in response to stimulation with variant-specific antigens, suggesting that these variants do not escape T-cell-mediated immunity [69]. In the case of the Delta variant, there was a 6-times lower neutralization activity compared to sera from individuals recovered from the wild-type infection [70]. The changes in the spike protein, particularly in the E484 protein residue, represent one of the main mechanisms resulting in escape from the immune system. The E484 substitution is estimated to have repeatedly emerged in the SARS-CoV-2 population [71]. Significant changes exhibiting a high degree of immune evasion were found in Omicron and its subvariants (BA.1, BA.2, BA.4, and BA.5) [72].

## 5. COVID-19 in the General Population

COVID-19 primarily affects the respiratory tract. Symptoms include cough, dyspnea, hemoptysis, pneumonia, and systemic symptoms, such as fever or fatigue. There are several non-respiratory manifestations, including diarrhea, loss of taste or smell, and a broad spectrum of neurological involvement, including ischemic or hemorrhagic stroke, dizziness, headache, altered mental state, Guillain–Barré syndrome, or acute necrotizing encephalopathy. Patients with mild symptoms usually recover after 1 week. However, severe cases may progress to respiratory failure due to alveolar damage as the leading cause of death [75,76]. A systemic review by da Rosa Mesquita et al. reported clinical manifestations from data from more than 100,000 patients. Fever and cough were reported in more than half of the patients. Dyspnea, fatigue, sputum hypersecretion, and anorexia were present in 20–30% of patients. Neurological involvement occurred in 20.8% of patients, and gastrointestinal symptoms, such as nausea/vomiting, diarrhea, or abdominal pain, in 9.6%, 7.3%, and 5%, respectively [77]. However, there are differences in clinical outcomes between virus variants. In particular, the Delta variant is associated with increased severity and prolonged viral shedding [78,79,80]. Patients infected with the delta variant of SARS-CoV-2 also show slower recovery [81]. In contrast, Omicron variants represent less-severe forms of COVID-19. Reduced severity has been reflected in almost all aspects, such as hospitalization, ICU admission, oxygen therapy requirement, or deaths [82,83,84,85], despite reduced efficacy of neutralizing activity of convalescent or post-vaccination antibodies against other virus variants and transmissibility [86]. Virus evolution also significantly affects the risk of re-infection [87].

From January 2020 to December 2021, 5.94 million deaths were reported worldwide; however, the estimated number may be even higher, reaching 18.2 million. The global all-age rate excess of mortality due to COVID-19 was 120/100,000 of the population. The highest excess was reported in South Asia, North Africa, the Middle East, and Eastern Europe. At the country level, the highest excess mortality rates were observed in Russia (375/100,000), Mexico (325), Brazil (187), and the USA (179) [88]. This results in a reduced life expectancy. While Western European countries experienced bounce-backs, Eastern Europe and the USA maintained substantial life-expectancy deficits in 2021 [89]. Booth et al. assessed population risk factors for severe disease and mortality in a meta-analysis. The authors included 76 studies with >17 million patients across 14 countries. They found that male sex (odds ratio 1.62), age above 65 years (odds ratio; OR 2.14–3.75), obesity (OR 1.8–2.02), smoking (OR 1.12–1.26), cardiovascular disease (OR 3.37), chronic lung diseases except for bronchial asthma (OR 3.54), active cancers (OR 3.19), immunosuppression (OR 1.17), and chronic kidney disease (OR 3.5) were the significant factors contributing to severe outcomes. The same factors were associated with increased mortality rates (Table 2) [90]. Interestingly, male sex predisposition for poorer outcomes of infections, including sepsis, has been described previously before the SARS-CoV-2 outbreak [91,92]. The differences may be explained by the diverse immune responses—higher CD4+ T-cell counts, more robust CD8+ T-cell cytotoxicity or increased production of immunoglobulins by B cells [93,94], and hormonal differences [95].

In a meta-analysis by Sunjaya et al., the risk of severe COVID-19 in patients with bronchial asthma was lower compared to non-asthmatic individuals. Additionally, no significant differences in hospitalization, ICU admission, or ventilator use were found between asthma and non-asthma groups [96]. The use of inhaled corticosteroids may explain this finding [97]. Choi et al. performed a systematic review of risk factors in children. They revealed neonate status (risk ratio 2.69), chronic lung disease (relative risk; RR 2.62), diabetes (RR 2.26), prematurity (RR 2.0), heart disease (RR 1.82), immunosuppression (RR 1.44), obesity (RR 1.43), and neurological diseases (RR 1.18) as the main factors [98]. Over 10% of patients reported signs and symptoms that developed during or after SARS-CoV-2 infection and persisted for more than 12 weeks. Post-acute sequelae (PAS) of SARS-CoV-2, also called post-covid syndrome (PCS), can affect a wide range of organs and systems. The underlying pathophysiological processes are still not entirely revealed. The persistence of SARS-CoV-2, reactivation of other viruses, persistent tissue damage, or immunity-triggered inflammation characterize the assumed candidate mechanisms. Inflammatory features, including type I and type III interferons, IL-6, and PTX3 molecules, are reflected in the findings of potential biomarkers of PAS. The most-common manifestations affect the lungs, neurocognitive, and cardiovascular functions [99,100]. In a systemic review by Michelen et al., weakness (41%), general malaise (33%), fatigue (31%), concentration impairment (26%), and breathlessness (25%) were reported. More than one-third of the patients complained about reduced quality of life [101]. Other observed symptoms include anosmia, hair loss, sneezing, and impaired sexual function. The risk of PAS increases with female sex, age >50 years, smoking, and the presence of comorbidities (psychiatric conditions, obesity, arterial hypertension, and immunosuppression) [102,103]. The stronger immune response and hormonal differences may explain hyperinflammatory status in females contributing to the long-term sequelae, in contrast to the favorable outcome of acute phases of COVID-19 described in female patients [94,95,104].

## 6. COVID-19 in Primary Antibody Deficiencies

Secondary immunodeficiencies, including immunosuppression, have been identified as a risk factor for severe outcomes of SARS-CoV-2 infection in the general population. This has raised a question regarding IEI and PADs, which represent the most-common primary immunodeficiency. However, the first report showed ambiguous results. Marcus et al. observed a minor impact of the infection in a small retrospective cohort of 20 patients with primary immunodeficiency from Israel. Eighty percent of patients (n = 16) had humoral immunodeficiency. Almost one-third (n = 6) of the patients were completely asymptomatic and were diagnosed following exposure to a positive-tested person. Sixty-five percent of patients (n = 13) had symptoms commonly associated with COVID-19. One patient experienced pneumonia. All patients were classified as having mild disease [105]. Similar results were reported by Goudouris et al., who assessed the course of the disease in 121 Brazilian patients in a multicenter cross-sectional study. The largest group, represented by 43.8% of patients (n = 53), was predominantly antibody-deficient. In most patients, the disease course was mild (54.5%, n = 66) or asymptomatic (17.4%, n = 21). Three patients died, three of whom had antibody deficiencies. Poorer outcomes were associated with older age and comorbidities rather than the type of IEI [106]. Comorbidities, as the major factor contributing to disease severity, were found in an international multicenter retrospective study by Meyts et al. The authors analyzed the data gathered from 94 patients with IEI. Fifty-six percent (n = 53) had PADs. Eleven percent of the patients (n = 10) had XLA, autosomal recessive agammaglobulinemia, and unclassified hypogammaglobulinemia. One-quarter of patients (n = 24) had mild disease. Sixty-three percent (n = 59) of patients required hospital admission. Almost all hospitalized patients with predominantly antibody deficiency were 45 years of age and older and two were above 75 years of age. Ten percent of the cohort’s patients (n = 9) died and five had antibody deficiency. All of them had pre-existing comorbidities, such as chronic cardiovascular disease, chronic lung disease, chronic kidney disease, diabetes, or lymphoproliferation, corresponding to the risk factors in the general population [107]. Shields et al. identified factors associated with poor prognosis in a cohort of 60 patients with IEI. These are mainly related to the primary underlying disease. A high risk of hospital admission was observed in patients taking prophylactic antibiotics, potentially reflecting chronic infection uncontrolled by immunoglobulin replacement therapy or more-severe immune deficiency. Lower baseline lymphocyte count is associated with mortality. In contrast to previous observations, the authors did not find a correlation between the prevalence of comorbidities and mortality. Only increasing age is associated with mortality risk. Inpatient mortality was greater among patients with CVID (61.5%, n = 8/13). Compared to the other PAD groups, only one out of four patients had unclassified primary antibody deficiency, and one out of three had polysaccharide antibody deficiency. No deaths were reported in the group of four patients with XLA. However, individuals with CVID were older, and a greater proportion was on immunoglobulin replacement therapy [108]. Milota et al. specifically examined the risk factors for hospital admissions in a nationwide retrospective study. The authors collected data from 81 patients with IEI and COVID-19. The largest proportion of infections occurred in patients with CVID (58%, n = 47/81), hereditary angioedema (19.8%, n = 16/81), and unclassified PADs. Asymptomatic infection was reported in 21% of patients (n = 17/81). Overall, 18.5% of patients (n = 15) required hospital admission, corresponding to a 2.3x higher risk (relative risk, RR) compared to the general population (7.9%). The highest number was among patients with CVID (21.3%, n = 10/47), corresponding to 2.68 RR. Eight patients required intensive care. Two patients died. This corresponded to a mortality rate of 2.4% in the entire IEI cohort and 13.3% among hospitalized patients. In contrast, the general population has reported a 1.7% mortality rate. The authors did not find statistically significant differences in the age or sex ratio of BMI between hospitalized and non-hospitalized patients. However, reduced lymphocytes, T cells, and NK cells significantly contributed to the risk of hospital admission and a lower number of B cells, IgA, and IgM hypogammaglobulinemia. There were no statistically significant differences in IgG trough levels in patients undergoing regular IRT. More than half of the hospitalized patients (60%, n = 9/15) had at least two comorbidities or IEI-associated complications, such as chronic lung disease (53.3%, n = 8/15) and cardiovascular diseases (33.3%, n = 5/15) [109]. Steiner et al. observed a robust T-cell response in patients with PADs with undetectable SARS-CoV-2-specific antibodies. The T-cell response in these patients was even stronger than that in healthy controls. The T-cell response was accompanied by increased markers of type I interferon response in most patients. Despite these findings, the authors observed prolonged viral shedding and a severe disease course, suggesting a critical role for humoral immunity in the immune response to SARS-CoV-2 [110]. Robust evidence of SARS-CoV-2 infection in patients with IEI was provided by the Italian national registry IPINet. Incidence, outcome, and risk factors were evaluated in a cohort of more than 3200 patients. The authors reported 131 cases of infection. Asymptomatic conditions were reported in 36.3% of pediatric patients and 24.5% of adult patients. Mild to moderate disease was reported in most patients (60.6%), followed by severe disease (3%). The overall infection fatality rate (IFR) was 3.81 compared to 3.28% in the general population. Severe disease and the highest mortality were seen particularly in CVID (IFR 4%), 22q11 deletion syndrome (IFR 8.3%), and Good syndrome (33.3%). In addition, PADs are associated with prolonged viral shedding. One-third of patients tested positive for SARS-CoV-2 for more than 3 weeks. Pre-existing comorbidities were described in only 6 of the 11 patients with severe COVID-19. However, no significant impact of lymphopenia and CD4+ T-cell lymphopenia on disease severity was found [111]. This overview summarizes the major studies that contribute to understanding SARS-CoV-2 in the context of IEI. However, several other studies have also been conducted [112,113,114,115,116]. The COVID-19 outcomes in patients with IEI are summarized in Table 3. 

Only sparse evidence investigating the outcome of COVID-19 in pediatric patients with IEI is available. The data suggest a worse outcome for IEI pediatric patients compared with non-IEI cohorts. In a study of 117 children with IEI by Moazzen et al., only nine cases had positive PCR results for SARS-CoV-2 within the follow-up period. The mean age was 47 months. All of them required hospital admission and intravenous antibiotics. Two children died—the patients with Omenn syndrome and combined immunodeficiency (CID) [117]. Both types along with primary immune regulatory and innate immunity disorders belong to the high-risk IEI groups associated with poor outcomes [118]. In a study by Abolhassani et al., a potential genetic defect was revealed in 28 patients, representing 90% of all pediatric patients enrolled in the study. The spectrum included the mutations affecting IFN signaling, T- and B-cell immunity, inflammasome, or complement system [119].

## 7. Conclusions

PADs are heterogeneous and comprise the largest group of IEIs. PADs are characterized by an increased susceptibility to infection and manifestations of immune-system dysregulation, leading to various noninfectious complications. In this context, patients with PADs may be considered as a risk population for a severe disease course of COVID-19 caused by SARS-CoV-2 infection. Several studies focused on the incidence, outcomes, and risk factors. Many studies reported ambiguous findings, probably reflecting demographic differences and virus evolution, leading to divergent virus variants with different features. These variants differ in infectiousness and severity. The main risk factors include older age, comorbidities, PAD-associated complications, and laboratory parameters, such as lymphopenia or hypogammaglobulinemia. Here, the humoral response seems to be a critical component in the immune system in controlling SARS-CoV-2 infection. This is probably the explanation for the poorer outcomes of PADs compared to other IEIs. Understanding COVID-19 pathogenesis and identifying risk factors may help us identify patients at high risk of severe COVID-19 in whom preventive measures, including anti-SARS-CoV-2-specific monoclonal antibodies or antiviral agents, can be introduced [120].

## Figures and Tables

**Figure 1 pathogens-12-00109-f001:**
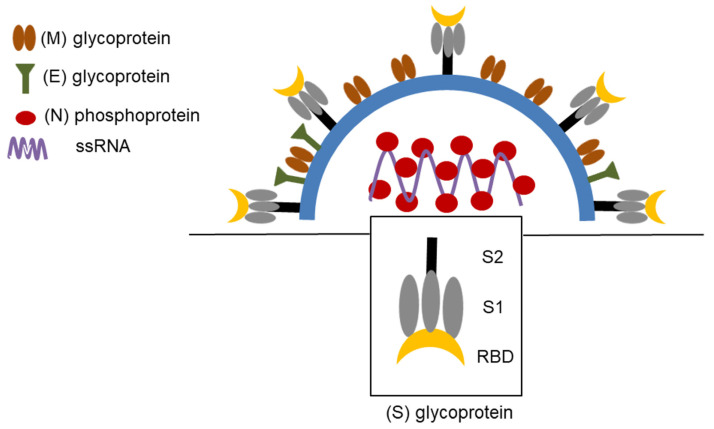
Structure of SARS-CoV-2 virus—Spike (S) glycoprotein, Membrane (M) glycoprotein, Envelope (E) glycoprotein, Nucleocapsid (N) phosphoprotein, single-stranded (ss) RNA, Receptor-binding Domain (RBD) [64].

**Figure 2 pathogens-12-00109-f002:**
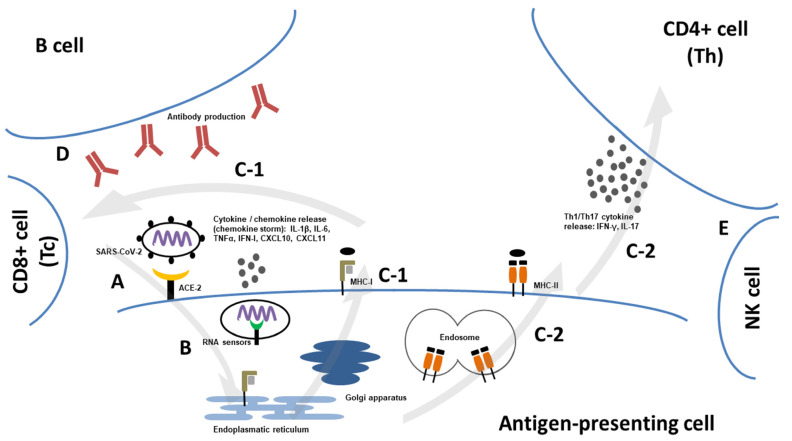
Pathogenesis of SARS-CoV-2—(**A**) virus entry mediated by Angiotensin-converting enzyme type 2 (ACE-2); (**B**) recognition of viral RNA by RNA-sensors (MDA5, RIG-1, Toll-like receptors 3/7/8) leading to inflammatory response with cytokine/chemokine release; (**C-1**) antigen presentation to and activation of cytotoxic C8+ T cells (Tc) via Major histocompatibility complex type 1 (MHC-I); (**C-2**) antigen presentation to and activation of CD4+ T-helper cells (Th) via Major histocompatibility complex type 2 (MHC-II); (**D**) activation of B cells and production of SARS-CoV-2-specific antibodies; (**E**) natural killer cell (NK) involvement [73,74].

**Table 1 pathogens-12-00109-t001:** Meta-analysis of the clinical manifestations of common variable immunodeficiency, pooled proportion (%) and 95% confidence intervals (95%CI) are displayed [37].

Manifestation	Pooled Proportion (%, 95%CI)
Chronic lung disease	44 [27,28,29,30,31,32,33,34,35,36,37,38,39,40,41,42,43,44,45,46,47,48,49,50,51,52,53,54,55,56,57,58,59,60,61,62,63]
Lymphadenopathy	30 (20–42)
Splenomegaly	29 (22–37)
Autoimmunity	27 (22–32)
Hepatomegaly	14 (8–22)
Enteropathy	9 (6–13)
Lymphoid malignancy	5 (3–7)
Gastric cancer	2 (1–4)

**Table 2 pathogens-12-00109-t002:** Meta-analysis of risk factors for COVID-19 associated with severe course and mortality. Odds ratio (OR) and 95% confidential intervals (95%CI) are shown (COPD, chronic obstructive pulmonary disease; CKD, chronic kidney disease [90].

Risk Factor	Severe (OR, 95%CI)	Mortality (OR, 95%CI)
Cardiovascular disease	3.37 (2.89–3.85)	4.04 (1.95–6.13)
COPD	2.47 (1.44–3.51)	2.68 (1.8–3.55)
CKD	3.5 (1.4–5.59)	2.79 (1.19–4.4)
Obesity	2.02 (1.02–3.01)	N/A
Higher age (>65)	2.14 (1.0–3.29)	1.89 (0.09–3.69)
Active smoking	1.22 (0.87–1.57)	2.13 (2.08–2.12)
Immunosuppression	1.17 (0.96–1.38)	2.31 (1.96–2.65)
Male sex	1.62 (1.29–1.94)	1.94 (1.51–2.37)

**Table 3 pathogens-12-00109-t003:** COVID-19 outcomes in patients with inborn errors of immunity.

Parameter	Ratio (%)	Reference
Asymptomatic infection	17–36.3	[105,106,111]
Risk of hospital admission	2.3	[109]
Infection fatality	3.8–20	[108,111]
Mortality	2.5–10	[106,107,109]

## Data Availability

Not applicable for this work.

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
