# Peer review of "Clinical Outcome of Coronavirus Disease 2019 in Patients with Primary Antibody Deficiencies"

_pathogens, 2023, doi:10.3390/pathogens12010109_

Round 1

Reviewer 1 Report

The manuscript entitled “Clinical outcome of coronavirus disease 2019 in patients with primary antibody deficiencies” reviews in part he evolution, structure and pathogenesis of Cov-19 virus. They have specifically focused on the disease outcome in patients with Primary antibody deficiencies along with co-morbidities in them. Authors have presented the data from several research papers where this study is done and presented them together in this review.

The abstract is precise and to the point. Overall, the review is written beautifully considering all the guidelines and present the importance of preventive measures for individuals at higher risk of severe Covid-19 infection.

The reviewer has no major comment but a few minor comments are there.

Minor Comments:

1)    Line 33; Comma should be there after older age instead of ‘and’.

2)    Line 52; Full stop is missing at the end of sentence. “However, disturbed…. immune dysregulation.”

3)    Line 58; Please rephrase- “The genetic…. agammaglobulinemia”. Meaning is not clear.

4)    Line 80-82; Please rephrase- “Additionally…. exhaustion”. Meaning is not clear.

5)    Line 82; ‘was’ should be replaced with ‘is’. The frequency of Tregs is negatively….

6)    Line 100; It feels like the two paragraphs are not linked. There was very sudden change from describing the problem to its treatment.

7)    Line 100-106; therapy should be replaced with treatment. Therapy is not he right word here. Also please tell a little bit more about the treatment options.

8)    Figure 1: Captions colors do not match with that in figure. Its confusing. Please match them.

9)    Line 167-168; Viral culture are negative after nine days but PCR is positive for 14-20 days. Why does the author think this is?

10) Line 179; C C reactive protein- ‘C’ is repeated twice.

11) Line 197; Please rephrase- “which prevent……. neutralization” Meaning is not clear.

12) Line 243; ‘was’ should be replaced with ‘were’.”5.94 million deaths were reported…

13) Line 249; The changes affected 29 countries…... Reviewer thinks this is not right number. Cov-10 epidemic affected many more countries. Please correct.

14) Line 254; ratio is misspelled as ‘ration’

15) Line 254; Why do you think male sex is more susceptible to female sex for cov-19. Can authors present some hypothesis or their views?

16) Line 263-264; Please rephrase- “Interestingly……. Group.” Meaning is not clear.

17) Line 283; Here it is written that risk is increased in female sex as opposed to previously mentioned in male sex, the risk is more. Please clarify. And if it is so, please explain why?

18) Line 292; n=80. How can value of ‘n’ be 80 in total study of 20 patients? Please correct.

19) Line 294; ‘percent is missing after sixty-five. ‘Sixty-five percent patients….’

20) Line 305; n=53. Please correct. In a study of 94 patients, 53% would be approximately 50.

21) Line 314; Authors have mentioned ‘PID’. Is it spelling error? If not please write the full form for the first time.

22) Line 340. Please write full form of ‘IRT’ when using first time.

23) References are missing at several places- Line187-188, 194-195, 201-203,223-227, 243-245 and 272-276.

24) Has there been any study done specifically on children with IEI and their susceptibility to disease?

25) Please make a table for meta- analysis to compare ‘OR’ or ‘RR’ in patients with IEI only and with comorbidities to better understand the risk. This will also summarize the review and what authors are trying to convey.

Reviewer 2 Report

The presented review on

Clinical Outcome of COVID-19 in Patients with Primary Antibody Deficiencies demonstrates a high scientific level and can be published in the journal Pathogens.

However, the manuscript needs to be improved:

Lines 287-290 - are we talking about primary or secondary immunodeficiencies?

In section 7, quantitative data must be checked:

Lines 290-292 - "Marcus et al. observed a minor impact of the infection in a small retrospective cohort of 20 patients with primary immunodeficiency from Israel. Eighty percent of patients (n= 80) had humoral immunodeficiency". - 80% of 20 = 80?

Lines 294-295 "Sixty-five patients (n= 13)" - percent?

Lines 304-305 - "The authors analyzed the data gathered from 94 patients with IEI. Fifty three percent (n= 53) had PADs" - the same.

Check other values too
